# On China's image constructed from western news coverage of China's humanitarian aid

Lei Sun[ID]*

Department of English, Foreign Language College, Tianjin Normal University, Tianjin, China

* sunlei003068@tjnu.edu.cn

## Abstract

Despite the growing body of research on media representations of China during the COVID-19 outbreak, limited scholarly attention has been devoted to Western news coverage of specific events involving China. Based on a self-built corpus of reports from English-speaking news sources on "China's Humannitarian Aid to Europe (2019-2024)", this study aims to contribute to the scholarship by examining how Western news media represents China's humanitarian aid to Europe, using quantitative and qualitative methods. The study employs corpus-based Critical Metaphor Analysis to analyze how China is linguistically represented through discursive strategies. Guided by the conceptual metaphor THE NATION AS PERSON, the analysis reveals a series of secondary metaphors—including CHINA AS PERSON, EUROPE AS PERSON, and AMERICA AS PERSON. In particular, CHINA AS PERSON is associated with multifaceted images such as "a politician," "an upstart," "a vendor," "an aggressor," and "a dictator," which collectively serve to undermine China's efforts and reframe humanitarian aid as a diplomatic maneuver. This highlights the effectiveness of corpus-assisted CMA in in analyzing media representations of national images amid a global health crisis.

## 1. Introduction

The COVID-19 pandemic, which emerged in 2020, has had profound implications for global governance. From March 12, 2020, the World Health Organization (WHO) declared COVID-19 a pandemic, to Octtober 23, 2024, WHO had reported over 777 million confirmed cases and more than 7 million deaths worldwide [1]. During this global emergency, humanitarian needs surged, particularly evident in the competition for medical supplies in March and April 2020 and in the UN's Global Humanitarian Response Plan for COVID-19. Since February 27, 2020, China has provided assistance to over 150 countries and several international organizations in the form of medical supplies, medical teams, and funding support [2]. In this study, the term "humanitarian aid" is used to refer to China's overseas medical and material

**Data availability statement:** All relevant data are within the manuscript and its Supporting Information files.

**Funding:** This work was supported by the Tianjin Education Commission Research Project (CN) [grant number 2024SK022]. The funders had no role in study design, data collection and analysis, decision to publish, or preparation of the manuscript.

**Competing interests:** The authors have declared that no competing interests exist.

assistance during the early stages of the COVID-19 pandemic, as it has more preference offically and academically, although other terms medical aid and medical assistance occasionally appear in media reports. China's image, particularly in the context of its humanitarian assistance to other countries, has attracted substantial international media attention, for the COVID-19 pandemic has not only posed global health threats but also intensified geopolitical narratives and media constructions of national identities.

To date, extensive research has examined Western media representations of China's image during the pandemic. However, the portrayal of China's humanitarian aid as a distinct event—with strategic goals such as enhancing geopolitical influence and reshaping global governance—has received less attention [3]. Other studies have examined China's medical aid to Africa, focusing on its diplomatic communications and soft power [4,5]. Moreover, research on China's vaccine diplomacy in the Middle East and North Africa (MENA) concludes that it has effectively positioned China as a "hero" in delivering health services to developing countries [6]. The provision of health assistance to the Pacific region has also been explored [7]. Some studies on China's aid to Europe suggest it serves to reinforce its established image [8], position itself as a responsible global leader [9], or present itself as a provider of global public goods [10].

Existing literature has investigated China's international image through lenses such as soft power, public diplomacy, and crisis communication. However, three major gaps remain underexplored: (1) a lack of corpus-based studies examining how China's humanitarian actions have been framed by Western media during the pandemic; (2) insufficient integration of both quantitative corpus techniques and qualitative metaphor analysis to investigate such framing; and (3) a limited focus on the ideological functions and cognitive implications embedded in the labeling of China's aid as "mask diplomacy".

This study addresses these gaps by employing a corpus-assisted Critical Metaphor Analysis (CMA) to explore how Western news media construct China's image in their coverage of medical assistance to Europe from 2020 to 2024. Consequently, the following research questions emerge as academically significant:

1. How did English-speaking news media report on China's humanitarian aid to Europe during the outbreak of the COVID-19 pandemic, and why was this assistance labeled as "mask diplomacy" in their framing narratives?

2. What ideological implications are embedded in the metaphorical representations of China across different Western media outlets?

3. What are the cognitive interpretations of the conceptual metaphor THE NATION AS PERSON in these news reports?

This study addresses these questions by reviewing the relevant literature and exploring the cognitive features of the conceptual metaphor THE NATION AS PERSON in English-language news. It aims to elucidate how China's medical aid efforts were narrated by recipient countries and other Western nations, thereby enhancing

scholarly understanding of the complexities of soft power dynamics and the construction of national images in global diplomacy.

## 2. Literature Review

### 2.1. Framing theory and western media's coverage of China

Framing theory examines how media influences the perception of events and issues by highlighting specific aspects over others, thereby shaping interpretation and decision-making [11,12]. The theory is generally divided into three components: frame building, the presence of frames in news messages, and the effects of framing on audiences [13]. This theory has been widely applied to analyze Western media's coverage of China's image. Research indicates that Western media often depict China in an unfavorable light, reinforcing "othering" as part of broader socio-political processes [14,15]. This tendency is particularly evident during crisis situations, such as the SARS outbreak in 2003, where China received negative press coverage compared to other countries like Vietnam [16]. Although Western media frequently utilize conflict, negativity, and human interest frames in their coverage of China, these portrayals are not uniformly negative [17]. For instance, Australian media often frame China's narrative within the discourses of the "China threat" and "Chinese influence" [18], and such negative portrayals have been linked to reduced travel intentions towards China [19]. Interestingly, the representation of China in Western media is not static; it reflects evolving geopolitical dynamics and China's growing global influence.

Further studies have noted that China's portrayal in Western media has entered an "age of uncertainty," characterized by inconsistent and divided representations [20]. A longitudinal analysis of transnational European newspapers indicates that the coverage of China's emergence as a great power has fluctuated over time, suggesting that Western media representations of China are both dynamic and multifaceted [21,22]. This framing reflects the intricate interplay between Western values and perceptions of China's development [23]. Moreover, in the context of the China-US trade war, strategic framing is more evident in coverage from countries directly involved in the conflict, underscoring the influence of geopolitical interests on framing choices [24]. The effects of Western media framing extend beyond politics and economics to impact sectors such as manufacturing [25] and tourism [26] Collectively, these studies suggest that Western media coverage of China, as a facet of its soft power, significantly influences international relations [27].

### 2.2. Conceptual metaphor theory, critical metaphor analysis and China's image research

Conceptual Metaphor Theory (CMT), first introduced by Lakoff and Johnson [28], posits that language involves a systematic mapping from a concrete domain (source domain) to an abstract domain (target domain). According to this theory, conceptual metaphors enable individuals to understand abstract concepts by drawing analogies with more concrete experiences, with relationships, attributes, and knowledge from the source domain being transferred to the target domain. Metaphors serve important linguistic and communicative functions,in classification of ontology, orientation, and structure, specifically in the war, architecture, journey, and human metaphors. In constructing a national image, media frequently employ the metaphor THE NATION AS PERSON, which attributes human characteristics and behaviors to nations. Lakoff points out that THE NATION AS PERSON is a powerful and explicit metaphorical concept in conceptualizing the international community. According to the types of human emotion, countries are divided into friendly, hostile, and rogue countries. This conceptual metaphor is rooted in the notion of national interests, suggesting that a nation's economic vitality and military strength are akin to human health and vigor [29]. In language, one core metaphor runs through a discourse that dominates several secondary metaphors or images. Subsequent scholars, including Fauconnier [30] and Kövecses [31], have further developed this framework.

While Conceptual Metaphor Theory offers valuable insights into metaphorical mappings, it remains predominantly descriptive. To address the critical dimension of metaphors as instruments of ideological influence and social power, this study draws upon Critical Discourse Analysis (CDA) as a foundational theoretical lens. CDA examines how discourse

reflects and reinforces social power relations, revealing hidden ideologies embedded within texts [32,33]. Unlike purely descriptive methods, CDA emphasizes uncovering implicit biases and ideological perspectives embedded within linguistic choices, making it particularly suitable for media discourse analysis.

Integrating CDA with CMT, Charteris-Black [34] proposed Critical Metaphor Analysis (CMA), a methodology designed to examine metaphors critically and cognitively. CMA retains CDA's critical stance on power relations and ideologies but specifically emphasizes metaphor as a salient discursive strategy. It typically follows three steps: identification, interpretation, and explanation, mirroring Fairclough's three-stage approach to discourse analysis [32]. CMA has since been applied to various fields, including political discourse, mainstream news media, business communication, and medical discourse [35]. Recent corpus-assisted CMA studies have shown that this approach effectively combines quantitative analysis with qualitative depth, revealing underlying ideologies and biases in media discourse [15,19,36].

Research indicates that media commonly use metaphors to shape public perceptions of China. Some scholars have used CMA to analyze construction of China's national image by the mainstream media in western countries [36,37]. In terms of the related research with COVID-19 epidemic, studies show that the use of metaphors like war, chess, and examination in Chinese news reports aims to evoke patriotism, reinforce national identity, and mobilize public cooperation during crises [38,39]. Also, the comparison of the use of four conceptual metaphors in Chinese vs. American media is studied to reveal different sociocultural context and ideological values [40]. At present, the existing scholarly achievement concentrates on exploring news presentations in terms of war metaphors, architectural metaphors, journey metaphors, body metaphors, and so on. Despite these studies, there remains a gap in the literature regarding the use of human metaphors in constructing China's image in relation to its humanitarian aid to Europe during the early stages of the pandemic.

### 2.3. Mask diplomacy

The concept of "mask diplomacy" emerged during the COVID-19 pandemic. It was first proposed by Wong [9] with his analysis of three features, particularly in relation to China's efforts to improve its global image and expand influence, with skepticism toward China's humanitarian aid. This terminology reflects broader historical and geopolitical narratives prevalent in Western discourse, notably the portrayal of China as a strategic and potentially threatening "Other".Then the concept has also been discussed in studies related with China's humanitarian aid to countries in Europe and the Middle East and North Africa (MENA) region as part of its Belt and Road Initiative [6,41]. One study indicates that China's mask diplomacy efforts significantly improved media tone and increased the reproduction of China's preferred narratives in recipient countries [42].

However, the majority of the existing scholarship has noted that western perceptions of China's global engagement are often characterized by suspicion, containment, and critical evaluations of China's intentions and actions [31,42]. Thus, the widespread adoption of "mask diplomacy" rather than more neutral terms like "humanitarian aid" or "medical assistance" signals an ideological framing that aligns with historical Cold War narratives, ongoing containment policies, and neo-Orientalist perspectives. Despite growing academic attention to geopolitical dimensions of mask diplomacy, few studies explicitly interrogate how this term functions discursively within Western media narratives. This gap highlights the necessity of critically analyzing the ideological underpinnings and cognitive implications embedded in the metaphorical use of "mask diplomacy", a core focus of the present study.

### 3. Methodology

This study adopts a corpus-based critical discourse analysis framework, integrating corpus linguistics tools with Critical Metaphor Analysis (CMA) to examine the construction of China's image in Western media reports on humanitarian aid during the COVID-19 pandemic.

## 3.1. Data

**3.1.1. Data collection.** This study utilizes the Factiva news database to select comprehensive and reputable Western English-language newspapers, including *The New York Times* in America, *The Guardian* in Britain, *Der Spiegel* in German, and *The Australian* in Australia. The selection of these newspapers was based on the following clearly defined criteria, reputation and influence, geographical representation and ideological diversity and ownership structure. The selection of newspapers was based on the following criteria, summarized **Table 1**.

These media outlets collectively represent diverse geographical areas (North America, Europe, and the Asia-Pacific region), enabling the exploration of potential regional differences in the Western media's portrayal of China. Each selected outlet varies in political stance, ranging from liberal or progressive to conservative, along with critical and investigative journalism. Moreover, their different ownership structures further enrich the analytical scope by allowing comparisons of how ownership and political stance influence media portrayals of China.

The data retrieval period was explicitly set from December 30, 2019 (the date when China officially issued "An Urgent Notice on the Treatment of Pneumonia of Unknown Cause") to June 1, 2024 (the date marking the commencement of data analysis for this research), providing comprehensive temporal coverage of the evolution of the COVID-19 narrative over approximately four-and-a-half years.

**3.1.2. Corpus construction.** Building upon the collected dataset, this study constructed two distinct corpora to facilitate both broad and focused analyses: the general Corpus CC and the specialized Sub-corpus CHAE. Firstly, initial retrieval was carried out. Articles published within the specified timeframe were retrieved from Factiva using the keywords "Coronavirus", "COVID-19", and "China" yielding an initial dataset of 9,783 articles.

Secondly, manual screening and validation was conducted. All retrieved articles were carefully screened to exclude irrelevant items such as duplicates, unrelated stories, or brief mentions of the keywords without substantive relevance. After this rigorous process 9,251 valid articles were retained, forming the "Corpus of Western English-speaking News Reports on COVID-19 in China" (Corpus CC).

Thirdly, Sub-corpus Identification was conducted. Within the broader corpus (Corpus CC), a subsequent targeted search was conducted using specific keywords ("China's emergency humanitarian aid", "China's humanitarian aid", "China's medical assistance", and "China's medical aid") to isolate articles specifically covering China's aid efforts. This search revealed that these activities were frequently labeled as "mask diplomacy". A focused keyword search for "mask diplomacy" initially identified 25 articles, of which 21 were validated after excluding coverage related to regions other than Europe. This subset forms the dedicated analytical corpus named the "Sub-corpus of Western English-speaking News Reports on China's Humanitarian Aid to Europe" (Sub-corpus CHAE).

## 3.2. Methods

This study employs both quantitative and qualitative methodologies. For the quantitative analysis, the corpus analysis toolkit AntConc 3.5.9 [43] was used on the Corpus CC to investigate keyword frequencies, collocation patterns, and the distribution of "mask diplomacy" references across the sources.

**Table 1. Overview of Newspaper Selection.**

| Newspaper | Region | Political orientation | Ownership structure | Rationale for selection |
|---|---|---|---|---|
| *The New York Times* | U.S. | Center-left | Family ownership | Leading U.S. newspaper with global influence |
| *The Guardian* | U.K. | Left-leaning | Trust ownership (Scott Trust) | High readership and editorial independence |
| *Der Spiegel* | Germany | Center-left/Investigative | Employee-owned | Rigorous international investigative journalism |
| *The Australian* | Australia | Conservative | Corporate ownership (News Corp) | Represents Asia-Pacific conservative media perspective |

For the qualitative analysis, applying Critical Metaphor Analysis (CMA) to Sub-corpus CHAE. The Metaphor Identification Procedure Vrije Universiteit (MIPVU) [44] was systematically used to identify metaphorical language based on comparisons of contextual and basic meanings.

### 3.3. Data analysis

Before conducting metaphor analysis, basic corpus statistics were compiled. The Corpus CC contained 9,251 news articles, while the Sub-corpus CHAE included 21 articles, specifically focusing on China's humanitarian aid narratives. These datasets provided the empirical foundation for subsequent metaphor identification.

Following the MIPVU guidelines, metaphorical expressions were identified through a structured interpretive process. Initially, all lexical units within the corpus were segmented and examined individually. Each unit's contextual meaning was determined based on its immediate linguistic environment. Subsequently, the basic, historically earlier and more concrete meaning of the unit was established, often with reference to established dictionaries. When a contrast between contextual and basic meanings was identified, and the contextual meaning could be understood via comparison to the basic meaning, the lexical unit was classified as metaphorical.

For instance, in *The Guardian* (April 10, 2020), the expression "Europe turned its back on Italy" was analyzed. The contextual meaning referred to the European Union's inadequate support for Italy during the early stages of the pandemic, whereas the basic meaning involved a person physically turning away from another. This semantic mapping illustrates the conceptual metaphor THE NATION AS PERSON, portraying Europe as a human agent capable of relational betrayal.

Through this analytic framework, underlying conceptual metaphors, such as THE NATION AS PERSON, were systematically identified and analyzed to reveal ideological patterns within the discourse.

## 4. Findings

This section presents the empirical results derived from corpus-based and metaphor discourse analysis. Quantitative results on reporting distribution and news topics are first introduced, followed by qualitative findings related to metaphorical constructions.

### 4.1. Reporting distribution over time

**Fig 1** shows a pronounced peak in coverage beginning on January 1, 2020, with the majority of the over 6,000 reports generated during the early stages of the pandemic. Coverage declined sharply after this initial surge, although some activity continued until June 1, 2024. **Fig 2** which focuses on a specific event within the pandemic, reveals a similar early spike with approximately 20 reports in early 2020, followed by only 2–3 reports in later stages, indicating a significant drop in sustained coverage. The Sub-corpus CHAE is considerably smaller in scope and duration, with most reports concentrated in early 2020. This sharp decline in coverage suggests that media attention on China's humanitarian aid to Europe was largely a short-term phenomenon.

### 4.2. The news topics

**Fig 3** and **Fig 4** indicate that the predominant topics in the news are the Novel Coronavirus and infectious diseases/epidemics, highlighting the global health implications of the pandemic. In contrast to the broader coverage in the Corpus CC, the Sub-corpus CHAE exhibits a slightly greater emphasis on political and diplomatic dimensions, with the top three to five topics pertaining primarily to politics and diplomacy.

### 4.3. Frequency and modifier analysis of "mask diplomacy"

Using AntConc 3.5.9 to search the corpus for "mask diplomacy" revealed 21 relevant reports—comprising 4 from The Guardian, 7 from *The New York Times*, 1 from *Der Spiegel*, and 6 from *The Australian*—with the term appearing a total of

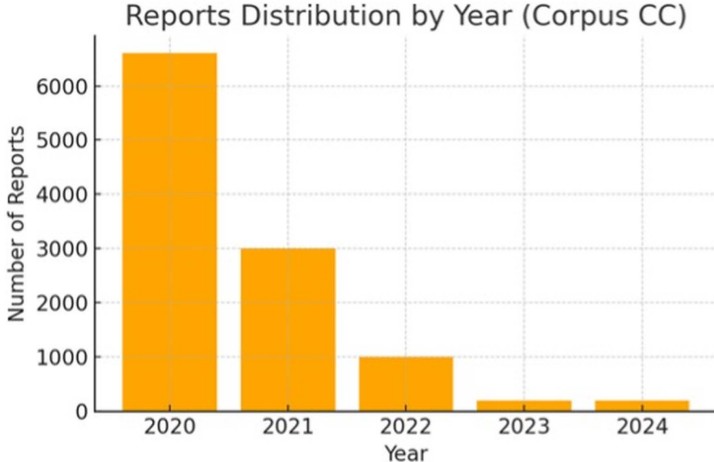

**Fig 1. Annual distribution of reports in the Corpus CC.**

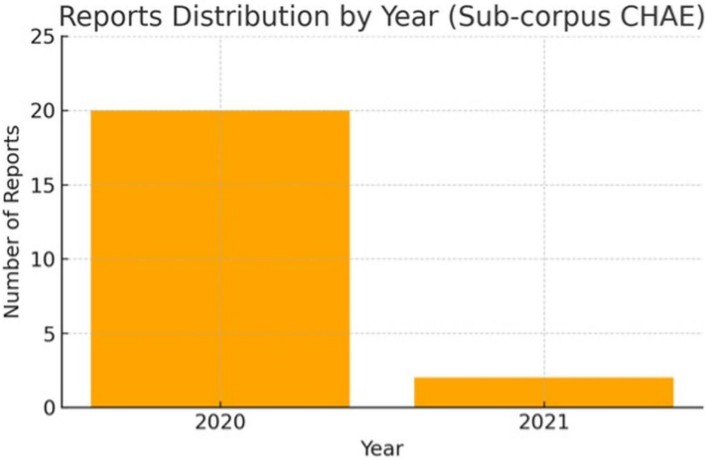

**Fig 2. Annual distribution of reports in the Sub-corpus CHAE.**

28 times. This terminology transforms the description of China's humanitarian aid into the self-coined term "mask diplomacy." Analysis of concordance lines indicates that modifiers denoting ownership (e.g., "China's," "Chinese," "Beijing's") appear 4 times; expressions such as "China's so-called" appear 7 times; and modifiers implying failure (e.g., "failure" and "fiasco") appear 3 times (Table 2).

In Table 2, terms like "China's" and "Chinese" might be neutral but their frequent use indicates a focus on China's agency, with a slight indication that the speaker's regonization of China's aid with its strong intention. The modifier "so-called" expresses the speaker's disapproving attitude, which reflects the skeptical or hostile attitude of these media outlets towards China's media assistance. Terms like "fiasco" and "failure" convey a strong negative evaluation, emphasizing perceived shortcomings and failures of China's "mask diplomacy". Using diction like "The failure of 'mask diplomacy'" in the reports 3 times, also illustrates the speaker's presupposition that the aid itself is doomed to fail with the negative and skeptical attitude.

Also, the distribution of these reports in the news section is 12 out of 21 articles are located in the international section, 3 out of 21 are in the nation section, 6 out of 21 are in the opinion section, 1 out of 21 are in the economy section,1 out of 21 are in the features.

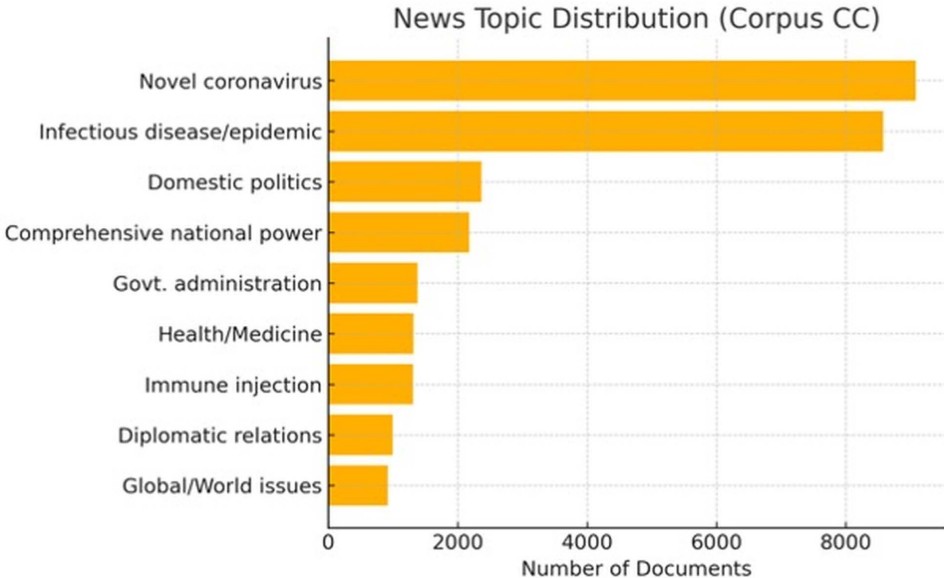

**Fig 3. Distribution of major news topics in the Corpus CC.**

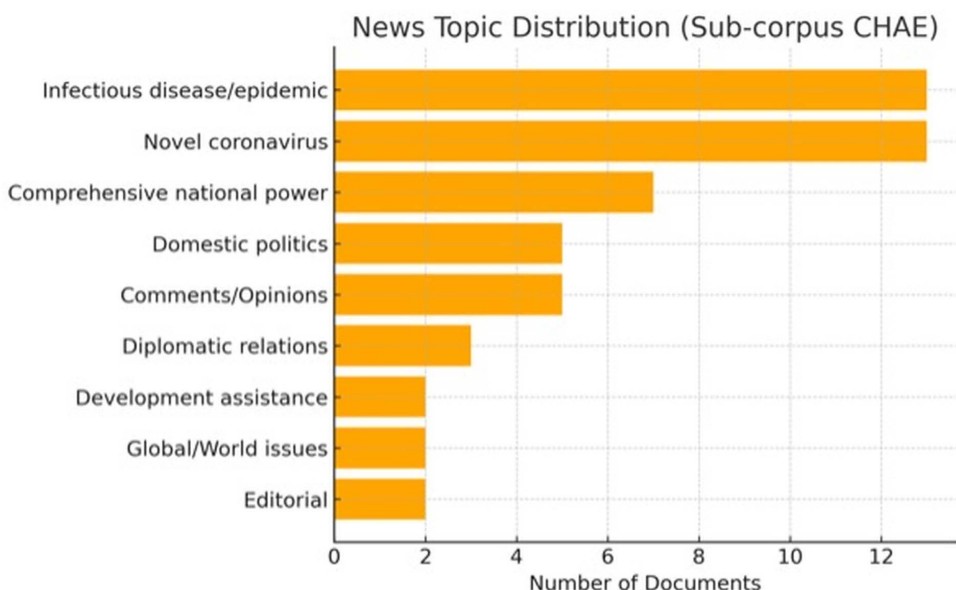

**Fig 4. Distribution of major news topics in the Sub-corpus CHAE.**

### 4.4. Idenfication of the Conceptual Metaphor of THE NATION AS PERSON

Through the lens of Critical Metaphor Analysis, metaphorical expressions within the corpus were identified using manual recognition with AntConc 3.5.9 and the MIPVU method (Steen et al., 2010). The identified metaphorical elements corresponding to THE NATION AS PERSON are summarized (**Table 3**).

**Table 2. Modifiers preceding "mask diplomacy" and their Frequencies.**

| The modifier before "mask diplomacy" | | Frequency /times | Source |
|---|---|---|---|
| Indicating ownership | China's | 4 | *The New York Times* 2020-07-01; *The Australian* 2020-03-28; *The Guardian* 2020-04-29; *The Guardian* 2021-03-14 |
| | Chinese | 1 | *The Australian* 2020-03-25 |
| | Beijing's | 1 | *Der Spiegel* 2020-08-02 |
| Indicating attitude | was dubbed | 1 | *The Guardian* 2020-07-24 |
| | are calling | 1 | *The Washington Times* 2020-04-11 |
| | Beijing's so called | 1 | *The Australian* 2020-05-29 |
| | be known as | 1 | *The Guardian* 2020-04-29 |
| | China's so called | 2 | *The New York Times* 2020-04-16; *The New York Times* 2020-11-9 |
| | Their much-vaunted | 1 | *The Australian* 2020-06-16 |
| Indicating failure | The fiasco of | 1 | *The Australian* 2020-05-05 |
| | Their failure at | 2 | *The New York Times* 2020-09-22 |

**Table 3. Metaphorical expressions in the Sub-corpus CHAE.**

| | Keywords on China's metaphor | Keywords on Europe's and US's metaphor |
|---|---|---|
| The noun as title | partner(5), rival(3), friend(2), pariah(4), hero(2), savior(1), leader(2), contender(1) | Europe: player(1), driver(1), friend(1) |
| Action noun | refusal(4), failure(4), claim(4), action(4), behest(2), effort(2), blitz(1), obfuscation(1), investment(1), insistence(1), punishment(1), ambition(1), betrayal(1), ability(1), aid(1), generosity (1), management(1), largesse(1) | Europe: sovereignty(6), solidarity(5), unity(3), resilience(1), approach(1), frustration(1), leadership(1), unity(1), cooperation(6), friendliness(2), alliance(1), kowtow(1) US: anger(1), goodwill(1) |
| Action verb | attempt(4), face(4), bully(4), pledge(3), lie(3), be willing to(3), send(3), ask(4), promote(3), pursue(3), expect(2), seek(2), use(2), compete(1), deny(1), complicate(1), mishandle(1), signal(1), view(1), work(1), encourage(1), hope(1), seek(1), undertake(1), want(1), confirm(1), emerge(1), employ(1), expect(1), help(1), hoard(1), indicate(1), lend(1), seek(1), inform(1), ally(1), violate(1), seize(1), promise(1), play down(1), aim(1), challenge (1), fan(1), threaten(1), reward(1), punish(1) | Europe: stand up(3), be capable of(2), sovereign(1), defend(1), assert(1), invite(1), mince(1), act(1), insist(1), allow(1), ask(1), beg(1), rely on(1), demonstrate(1), combine(1), beg(1), be locked (1) US: threaten(1), sue(1), waylay(1), block(1), be accused of(1) |
| | Source dormain resonance value 70*123 | Source dormain resonance value 39*59 |

From the table above, it can be seen that in the core conceptual metaphor of THE NATION AS PERSON, western media forms a series of secondary metaphors, such as CHINA AS PERSON, EUROPE AS PERSON and AMERICA AS PERSON.

## 4.5. Cognitive interpretation of THE NATION AS PERSON

Table 3 indicates a strong association between the conceptual metaphor THE NATION AS PERSON and the term "mask diplomacy." Within the "noun as title" category, China is characterized by both positive metaphors (e.g., partner, friend, hero, savior, leader) and negative ones (e.g., rival, pariah, contender), reflecting a dual and complex portrayal of China's role during the early COVID-19 outbreak. Action nouns associated with China (refusal, failure, claim, etc.) emphasize conflict, effort, and ambition, indicating their perception of dynamic and often contentious engagement with the world. The

verbs (attempt, face, bully, etc.) further highlight active, often aggressive interactions, portraying China as both an initiator and responder in the early outbreak. In contrast, Europe and the US are depicted with more defined roles: Europe is characterized with terms like "player," "driver," and "friend," reflecting a cohesive and proactive identity, whereas the US is represented with fewer and more confrontational metaphors (e.g., threaten, sue, waylay, block).

These findings suggest that Western media portray Europe with a more unified and less contentious identity compared to China, which is depicted in a multifaceted and often ambivalent manner. The following qualitative analysis further examines the cognitive interpretations of THE NATION AS PERSON.

(1) China is a manipulative politician.

**Example 1.** But Chinas mask diplomacy has fanned a mistrust of the EU kindled years ago by Italy's populists. The legislator sounded despondent."I wish I could say this mask diplomacy isn't working very well. But it is, unfortunately."(*The Guardian*, 2020-04-29)

This report titled "How the face mask became the world's most coveted commodity", sparked a debate in British Media about masks and mask diplomacy. The idiom "fan the flames of something" implies inciting negative emotions(e.g., hatred or distrust). By using "fan", *The Guardian* implies that China's assistance escalates distrust and hatred among European countries. The quoting of the unnamed Italian coalition government legislator intends to be neutral and objective. However, its seemingly neutral acknowledgement conveys strong anxiety about China's aggressive diplomacy. In this report, *The Guardian* indicates that China is a politician sowing discord between EU countries for its own selfish gains and coveys their anxiety about it.

**Example 2.** Relations with Europe also have taken a turn for the worse, precipitated by China's ill-conceived attempts to score a propaganda win by supplying poor quality face masks and other medical equipment to countries afflicted by the coronavirus. Chinese officials compounded the fiasco of their mask diplomacy by faulting Europeans for their allegedly poor crisis management. (*The Australian*, 2020-06-16)

"Ill-conceived attempts" and "a propaganda win" are the behavior used by *The Australian* to metaphorize China with intense political intention. Furthermore, it frames China's criticism of the European poor crisis management "compound the fiasco of" mask diplomacy. "Compound" usually followed in the collocation by nouns with negative nouns, indicating the worsening situation. "Fiasco" connotates failure with a strong sense of shame. Using the affirmative form "can fault sb" indicates that Australian media suggests that China's mask diplomacy is a complete failure and its government tries to make European countries, which are innocent victims, take the blame. *The Australian* constitutes the source domain through "ill-conceived attempts","a propaganda win", and "faulting Europeans", mapping to the target domain that China is an irresponsible politician who promotes political propaganda and is doomed to failure.

**Example 3.** The campaign was not all punitive, though; it also included incentives for good behavior. One facet of the response was "mask diplomacy": wielding China's near-monopoly over essential P.P.E. manufacturing as a tool for rewarding friends and punishing perceived enemies. (*The New York Times*, 2021-7-11)

In the report "Voice of China", "rewarding friends" and "punishing perceived enemies" are the representative metaphor for THE NATION AS PERSON. The action verbs of "reward" and "punish" are set from the source domain to map the target domain, in the cognitive process of conceptualizing China's image. In addition, the attribute of the enemies "perceived" maps China into the target domain with an image of a politician with a scheme. The binary opposition of human actions of "reward friends" and "punish enemies" reinforces the conceptualization of "mask diplomacy" as a tool to manipulate other nations.

(2) China is an authoritarian dictator.

**Example 4.** China is also a systemic rival, however, and it is increasingly going on the offensive, also vis-a-vis Europe. Beijing's "mask diplomacy" coupled with a disinformation campaign in the midst of the coronavirus crisis is just one current example. The leadership of the authoritarian, one-party state passes up no opportunity to drive a wedge between the EU

member states and weaken them. We are locked in a tough competition of values stemming from very different concepts of society. (*Der Spiegel,* 2020-08-02)

This report "The Security of Our Citizens Is at Stake" begins with a discussion of the complex and dynamic relationship between China and the EU, in which China is both a partner and a competitor. It suggests that a win-win result can only be achieved through bilateral cooperation. However, from this perspective, China is metahporized into "a systemic rival" as an imaginary enemy. Through its collocation with "mask diplomacy", the public is mapped to associate China's humanitarian aid with "disinformation". It employs the verb phrase "to drive a wedge" to indicate its sowing discorded behavior. With the subject "authoritarian", China is mapped as an ill-intentioned and dictatorial person. The use of the verb "be locked" depicts the EU in innocent passiveness caught in a complex competition of the clashing values, while its passive and innocent role works as a foil to emphasize China's dictatorship.

(3) China is a dishonest vendor with inferior products.

**Example 5.** European frustrations with Chinese policies have been mounting, but they crystallized this year in the wake of the coronavirus pandemic. China's obfuscation of its early missteps in containing the coronavirus and its failure at "mask diplomacy" soured public sentiment in several countries, especially the Netherlands and Spain, where protective gear and other supplies that were purchased, not donated, were found to be defective. (*The New York Times,*2020-09-22)

In this report titled "China, Seeking a Friend in Europe, Finds Rising Anger and Frustration", it depicts the term "obfuscation" in denotation of "deliberatelly". Thus, China is personified as a person blaming others with "obfuscation", creating the criticism against others deliberately. Through the combination of this action and the following phrase China's "failure at mask diplomacy", this report intends to construct the blame to China. Meanwhile, it frames linguistically with the attributive clause to illustrate that most of the Chinese masks brought by the Netherlands and Spain are in poor quality, associating it to the stereotype of "made in China" products of low quality. Hence, this report attempts to map China an irresponsible person at first, and then a tricky person preferring to muddle through trouble. Finally, it guides the public to associate with the stereotypical image of China, sending the audience a signal of being wary of China.

(4) China is an upstart with great fortune.

**Example 6.** Closer to home, the Philippines, Indonesia and Malaysia also have accepted Chinese largesse. But leaders such as Serbia's President Aleksandar Vucic, who went to the airport to welcome Chinese supplies, gushing about his belief in "my brother and friend Xi Jinping", need to be circumspect. China's mask diplomacy is an attempt at what the Lowy Institute's Richard McGregor termed "the fastest turnaround in global history from pariah to hero".(*The Australian*, 2020-03-28)

In the report titled "Coronavirus ignites epic battle of the superpowers", Australian media discusses the rise of China's international status in the "mask diplomacy" event. The term "largesse" is used to denote charitable giving, yet it is framed as an upstart's lavish handout intended to win favor, which is met with ridicule. The verb "gush" is employed, connoting both exaggerated admiration and disapproval, particularly in reference to the Serbian president's amiable response towards China. Moreover, the use of "attempt" forms a semantic correlation with the direct quotation of McGregor, a researcher at the Lowy Institute, who defines China's humanitarian aid as "the fastest turnaround in global history from pariah to hero". McGregor's use of "pariah" and "hero" to map China's international status is full of mockery and skepticism, which reinforces the use of "attempt" in negative connotation. Through cognitive mapping, this report attempts to construct an image of China as an upstart with money to cover mistakes, in vain through.

(5) China is an ambitious aggressor.

**Example 7.** This spring, Beijing energetically promoted its exports and overseas donations of medical supplies and asked foreign politicians to thank China publicly for the shipments. …The study, however, cast doubt on whether the humanitarian aid blitz really took place, since China's exports were down in March from a year earlier. (*The New York Times,* 2020-05-06)

In this report, the US media metaphorizes China with human behavior through expressions "promote energetically" and "ask foreign politicians to thank China", indicating China has an ulterior motive with malicious intentions. "Blitz" with the connotation of the sudden attack, especially after WWII, it was closely related to the heavy bombing to British cities by German aircraft in 1940, with a strong negative connotation with inhumanness and injustice. In this way, *The New York Times* narrates China as an unfavorable enemy image through the indication of this assistance to unjust German raids.

Also, it employs an oxymoron with the attributive "humanitarian aid' to modify the negative noun "blitz", in the tone of suspicion, making a cognative correlation between China and Germany. Using the metaphorical rhetoric of "blitz", China's humanitarian aid is depicted as an inhuman and futile political strategy, and China has been mapped to an aggressor.

(6) Europe is an innocent victim; America is an irritated hothead.

**Example 8.** Alluding to those who would play the continent off the world's two quarreling powers, he (refers to Charles Michel, the President of European Council) added, "Europe needs to be a player, not a playing field." (*The New York Times,* 2020-09-22)

In this report titled "China, Seeking a Friend in Europe, Finds Rising Anger and Frustration", it states that China and the US are now in a state of parity. Although the previous paragraph has hinted that China's "mask diplomacy" would work, in this paragraph US media swithces to negative attitude. Quoting the statement of the President of the European Council, in which Europe is personified as a player, US media hopes Europe is a "player" instead of a "playing field" in the dynamic international relationship. Here, the metaphor of "player" highlights the initiative of EU as person competent to make his voice heard, while the "playing field" does not. *The New York Times* depicts Europe as an innocent victim without intention of participating in the global governance competition, but is passively involved in it.

(7) Europe is a knight fighting for justice.

**Example 9.** With the largest multi-annual financial framework in its history and further coronavirus aid totaling 750 billion euros, the EU is now setting the course for the European future. A Chinese proverb tells us that "It is better to be envied than pitied." A strong and sovereign Europe in a spirit of solidarity that protects its citizens and stands up as one and with determination for its values and interests in the world is a form of life insurance to be envied. This is a question of sovereignty, both vis-a-vis China and others. The good news is that we are in the driver's seat here.(*Der Spiegel*, 2020-08-02)

*Der Spiegel* first analyzes the political and economic competition between China and the US in Europe, believing that the epidemic is a crucial moment for Europe's development. In this paragraph, German media quotes the Chinese proverb "宁被人妒, 不受人怜", to metaphorize Europe into a heroic knight overcoming difficulties and fighting for the common values and interests of all Europes through the words of "protect", "stand up" and "solidarity." Despite of its misinterpretation of this Chinese proverb, it attempts to narrate Europe with efforts to justify their solidarity. The use of "in the driver's seat here" indicates that Europe takes everything under control as a driver grasping the steering wheel in global governance.

In summary, based on the qualitative research from Table 3 and the quantitative analysis, it can be seen that in the core conceptual metaphor of THE NATION AS PERSON, these media representations form a series of secondary metaphors, such as CHINA AS PERSON, EUROPE AS PERSON and AMERICA AS PERSON whose source domain is PERSON and target domain is CHINA, EUROPE and AMERICA In particular, CHINA AS PERSON has been mapped to multifaceted images. To understand the corresponding relationships in their conceptual structure, Fig 5 displays the mapping process THE NATION AS PERSON in detail.

## 5. Discussion and conclusions

### 5.1. Lexical framing of "mask diplomacy" in western media

The conceptual metaphor THE NATION AS PERSON emerged prominently in Western media discourse, particularly through the systematic use of specific linguistic expressions and metaphorical constructs. Table 2 reveals that the modifiers and collocations associated with "mask diplomacy" underscore frequent references to China, predominantly using

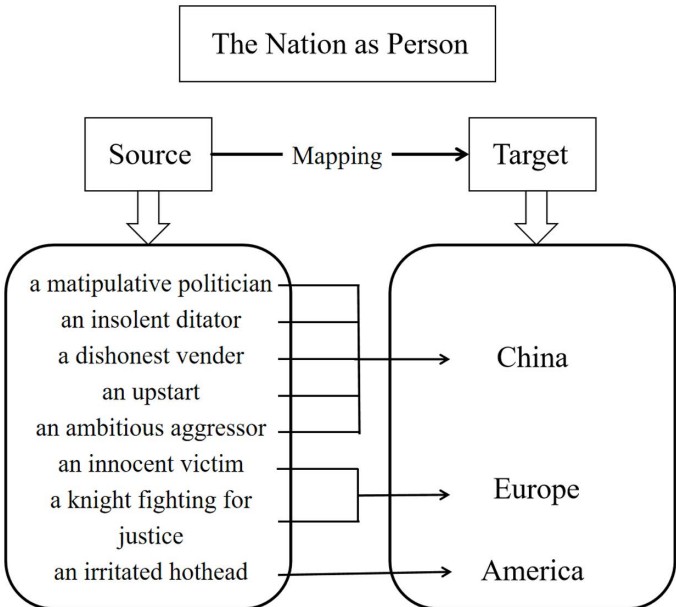

**Fig 5. The mapping of THE NATION AS PERSON.**

negative evaluative language. Moreover, the placement of these articles in the international section further emphasizes the geopolitical significance attributed to China's actions. These metaphorical framings correspond closely with Lakoff and Johnson's [28] foundational theory of metaphor, which suggests that metaphors structure our cognitive understanding of complex political realities. Charteris-Black [34] further emphasizes that metaphorical language in political and media discourses functions peruasively, framing how readers interpret nation-states and international actors. This strategic linguistic framing in line with Fairclough's [45] observation that media discourses actively construct, rather than neutrally reflect, social and geopolitical realities. Therefore, the labeling of China's aid as "mask diplomacy" suggests that it is perceived as part of a broader diplomatic strategy rather than purely humanitarian assistance.

Corpus-based quantitative and qualitative analyses confirm that these metaphorical representations consistently characterize China in negatively evaluative terms. Notably, China's humanitarian aid is frequently recast as a strategic maneuver rather than a benevolent gesture, underscoring the prevalence of evaluative negativity in Western discourse.

Building on previous research such as Kowalski [41], who emphasized China's strategic intentions, and Fahrisa et al. [6], who examined its health diplomacy functions, this study enriches their findings by approaching "mask diplomacy" from a discourse-linguistic perspective. It also supplements Müller et al. [42] by providing empirical corpus-based evidence of how the term is framed in Western media. In doing so, this research responds to and extends existing scholarship through a new analytical lens focused on lexical evaluation and metaphorical construction.

### 5.2. China's image in western media's discursive strategies in THE NATION AS PERSON

Based on both qualitative and quantitative analyses, Western news outlets employ the conceptual metaphor THE NATION AS PERSON to construct a series of images (**Fig 6**). These include portrayals of China as "a manipulative politician," "an upstart with great fortune," "a dishonest vendor with inferior products," "an ambitious aggressor," and "an insolent dictator." In contrast, Europe and America are framed as "an innocent victim" (Europe), "a knight fighting for justice" (Europe), and "an irritated hothead" (America), respectively, reinforcing China's constructed image. These images are accompanied by corresponding emotional connotations: the portrayal of China as a "dishonest vendor" evokes Eurocentric arrogance; as a

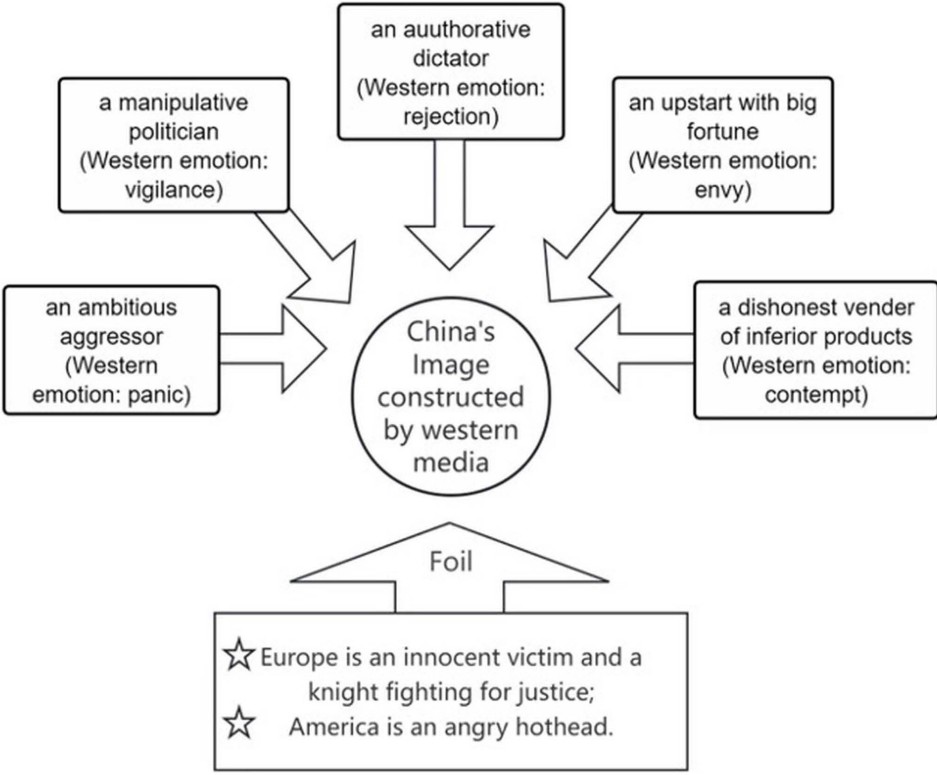

**Fig 6. The Cognitive framing of China's image.**

"manipulative politician," it inspires vigilance; and as a "dictator" or "aggressor," it elicits feelings of rejection and panic. By constructing parallel human images—depicting America as reckless and Europe as compromising—the media convey a guarded stance towards China's rising status. Notably, the cognitive framing of China as "the aggressor" is informed by a Cold War mentality, further reinforced by the portrayal of Europe as a knight fighting for justice.

This findings further extend existing scholarship on metaphorical framings. Whereas earlier studies have primarily focused on war, architecture, journey, or body metaphors [26,39], this analysis specifically addresses the previously under-examined use of human metaphors. By explicitly attributing human traits to nations, these metaphors carry potent ideological undertones, reflecting and reinforcing particular geopolitical narratives. Furthermore, as Zhang and Lin [39] argue, metaphors in crisis discourse carry emotive and cognitive resonance that shapes readers' orientation toward global actors. In this context, the portrayal of China through humanized metaphors reinforces pre-existing assumptions about its global intentions. This study contributes to this line of inquiry by demonstrating that metaphorical representation of China is not random but deliberate, operating through consistent patterns of lexical framing.

### 5.3. Ideological implication of THE NATION AS PERSON

The primary aim of this study is to examine how Western media discursively construct the notion of "mask diplomacy," yet it is also important to acknowledge the political significance embedded in the term. The consistent use of "mask diplomacy" instead of more neutral phrases like "humanitarian aid" reflects not only lexical preference, but also ideological positioning. Fairclough [32] argues that lexical choices in media discourse actively construct and sustain ideological frameworks, suggesting that the media's persistent labeling of China's aid as "mask diplomacy" strategically in line with Western

geopolitical skepticism toward China. In global discourse, humanitarian aid is increasingly perceived as an instrument of soft power and symbolic diplomacy. As previous studies have shown [41,42], China's pandemic-time medical outreach was widely interpreted as an effort to rehabilitate its international image and project itself as a benevolent global leader. Western media's metaphorical framing of this initiative, therefore, may not be ideologically neutral, but rather a reflection of deeper geopolitical anxieties and skepticism surrounding China's global rise.

Metaphors in news discourse serve as powerful tools to construct national images by attributing human characteristics to countries. Charteris-Black [34] elaborates that metaphorical representations in political discourse strategically reinforce underlying ideological positions. In the context of Western coverage, such metaphors often mirror broader geopolitical tensions and cultural attitudes. China is portrayed through multiple personified identities: as "a manipulative politician", it is depicted as strategically deceptive; as "an upstart with great fortune," it evokes suspicion and perceived unworthiness; and as "a dishonest vendor with inferior products," it reinforces stereotypical narratives around product quality. Other framings include "an ambitious aggressor" and "an authoritarian dictator," both of which conjure Cold War-era anxieties and human rights concerns. These images construct China as a multifaceted but ultimately threatening figure. In contrast, Europe is framed either as "an innocent victim" or as "a knight fighting for justice", embodying moral authority and unity, while the United States is often portrayed as "an irritated hothead", impulsive and defensive. These contrastive metaphorical constructions not only reinforce Western ideological positions, but also serve to rationalize skepticism or containment strategies toward China.

Fig 4 finds the western media's reports surge in the early 2020, then it did not maintain significant media presence afterward. This short-term surge can, thus, be indicated as a sign of China's humanitarian aid shattling the defense mechanism of some western countries, and challenges the Eurocentric ideology. Recent empirical findings [15,36,39] confirm similar patterns, demonstrating that Western media portrayals of China fluctuate significantly in response to perceived geopolitical threats or ideological challenges. The Othered China offered assistance while EU was in disarray and US is discredited in gloabal governance, which, to some extent, challenges the Cold War mentality. Such media portrayals, as Müller et al. [42] have argued, reflect deep-rooted geopolitical biases, reinforcing established ideological binaries and anxieties about China's global ascendancy. In a word, the constructing of a series of multifaceted images of China reflects the complex and nuanced sentiments of western news outlets, including the vigilance against "a politician", the rejection against "a dictator", the contempt over "a vender", the envy of "an upstart" and the panic about "an aggressor".

These metaphorical framings resonate deeply with broader Western ideological constructs and historical narratives. Depictions such as China as "an authoritarian dictator" or "an ambitious aggressor" reflect enduring Cold War mentalities, highlighting Western anxieties toward non-democratic powers. Similarly, the portrayal of China as "a dishonest vendor" draws upon historically entrenched, neo-Orientalist stereotypes that emphasize economic suspicion and moral inferiority. Such ideological stereotypes, as explained by van Dijk [33], serve to establish and perpetuate an "us-versus-them" dichotomy inherent in Orientalist discourse, thereby legitimizing geopolitical containment strategies and pervasive skepticism toward China's international initiatives. Thus, Western media's strategic metaphorical framing during the pandemic illustrates how language simultaneously invokes and perpetuates deep-seated ideological and cultural narratives [41,46,47].

While existing studies have addressed the geopolitical implications of China's humanitarian actions during the COVID-19 pandemic from the perspective of international relations and soft power diplomacy [47], this study contributes a complementary lens by offering a discourse-level, metaphor-focused media analysis. It supplements Lu and Yu's [40] findings, the distinct metaphorical portrayals reflect deeper ideological contrasts and sociocultural contexts, highlighting Western media's implicit geopolitical biases during the gloabal crises. By tracing how different Western media outlets metaphorically construct China as various kinds of human agents this research reveals the cognitive strategies and ideological functions underlying seemingly neutral humanitarian narratives. Such metaphorical framings not only shape public perception but also reflect broader anxieties regarding China's rising global status and contested image in Western discourse.

## 5.4. Media outlet's positionality and its framing of nation's image

Notably, Western media outlets exhibit nuanced variations in their framing of the conceptual metaphor PERSON, reflecting their diverse positions on China's image. The portrayal of China's image in international media is often shaped by a complex interplay of factors, including the political stance, ownership structure, and target audience of individual media outlets. Each of these elements can influence how China is represented, reflecting broader ideological, economic, and cultural dynamics. These factors shape the narratives and metaphors employed, such as CHINA AS POLITICIAN, resulting in diverse representations across different publications.

For the political stance, *The New York Times* is generally perceived as having a center-left bias, often endorsing liberal policies and Democratic candidates. It is owned by The New York Times Company, with the Ochs-Sulzberger family maintaining control through a dual-class share structure that ensures editorial independence. It aims to cater to an urban, educated readership, with a significant portion interested in in-depth political and international coverage. In its coverage, it employs a nuanced approach, using sarcasm to critique European countries' "flattering" attitudes toward China, suggesting such praise compromises their dignity. Employing dictions of "rewarding friends" and "punishing perceived enemies" characterize China's political strategies, blending irony with a modicum of respect, which reflects the *The New York Times*'s commitment to critical analysis, in line with its liberal perspective and intellectually engaged audience. In contrast, *The Guardian*, is known for its left-leaning orientation, advocating progressive and liberal viewpoints. Owned by the Guardian Media Group, which is in turn owned by the Scott Trust, *The Guardian* operates under a unique structure that ensures editorial independence and freedom from commercial pressures. Its readership is diverse, sharing a passion for insightful journalism, progressive values, and thought-provoking content. In framing CHINA AS POLITICIAN, *The Guardian* narrates China as a culpable politician responsible for fostering mistrust within the EU, using expressions such as "unfortunately, the mask diplomacy works" to convey regret and distress. Meanwhile, *The Australian* is known for its conservative orientation, often supporting right-leaning policies and perspectives. In framing CHINA AS POLITICIAN, *The Australian* adopts an overtly derogatory tone, using phrases like "ill-conceived attempts" and "a propaganda win" to directly map China as a manipulative politician, describing its actions as "a fiasco." This framing reflects its conservative stance and resonates with readers skeptical of China's global intentions.Thus, while the conceptual metaphor CHINA AS POLITICIAN is employed across Western media, each outlet's portrayal varies according to its ideological stance and intended audience, with American media distancing themselves by positioning Europe as a victim, European media expressing ambivalence, and Australian media demonstrating strong opposition to China's actions. Therefore, the political alignment of a media outlet plays a crucial role in shaping its coverage of China. Media organizations with distinct ideological leanings—whether liberal, conservative, or neutral—are likely to interpret and present information about China in ways catering to their political audiences. As media organizations continue to navigate complex geopolitical and economic realities, the portrayal of China will remain subject to the intricate dynamics of political ideology, ownership interests, and audience expectations.

The findings presented above explicitly illustrate the frame-building process articulated in framing theory [11,12,42]. Consistent with previous research highlighting the use of conflict and negativity frames [17,18], *The Australian*'s portrayal illustrates the "China threat" narrative, driven by its conservative ideological stance and corporate ownership, and *The Guardian*'s ambivalent yet critical stance underscores how editorial independence allows nuanced framing, resonating with its progressive audience. This results support previous framing research by highlighting that media representations of China reflect diverse, strategic editorial choices shaped by complex interactions of ideological, economic, and socio-cultural factors [20,22,24]. While prior studies have broadly noted ideological divergence in media framing of China (e.g., [17,20,22,24]), this study contributes a finer-grained, metaphor-oriented comparative analysis across outlets, highlighting how the same metaphor (CHINA AS POLITICIAN) is contextually reconfigured.

 

## 5.4. Limitations

Although the relatively small sample size of the Sub-corpus CHAE limits the generalizability of the findings, it remains sufficient for conducting an in-depth qualitative analysis of metaphorical and ideological discourse. Additionally, this study treats "Western media" as a relatively unified category, despite the fact that media outlets in different Western countries may adopt divergent perspectives and framing strategies when reporting on China.

Moreover, the exclusive use of English-language sources may overlook alternative representations found in non-English media. Future research could broaden the linguistic and regional scope to capture a more diverse range of perspectives.

Finally, while this study focuses on media framing, it does not explore China's medical aid as a diplomatic strategy. Future studies may examine how humanitarian assistance functions as a tool of soft power and how it is received in different geopolitical contexts.

## Supporting information

**S1. Data_File.**
(XLSX)

**S2. Coding_Scheme.**
(DOCX)

**S3. Table 1_Modifiers.**
(CSV)

**S4. Example_Quotes.**
(DOCX)

## Author contributions

**Writing – original draft:** Lei Sun.

**Writing – review & editing:** Lei Sun.

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
