## [Decision Letter · Decision Letter 0]

Apr 06 2025

Dear Dr. Sun,

We look forward to receiving your revised manuscript.

Kind regards,

Rafael Galvão de Almeida, PhD.

Academic Editor

PLOS ONE

“Tianjin Education Commission, China.”

“This work was supported by Tianjin Education Commission Research Project, China [grant number 2024SK022];”

“Tianjin Education Commission, China.”

Reviewers' comments:

Reviewer's Responses to Questions

**Comments to the Author**

1. Is the manuscript technically sound, and do the data support the conclusions?

Reviewer #1: No

Reviewer #2: Yes

2. Has the statistical analysis been performed appropriately and rigorously?

Reviewer #1: N/A

Reviewer #2: Yes

3. Have the authors made all data underlying the findings in their manuscript fully available?

Reviewer #1: Yes

Reviewer #2: Yes

4. Is the manuscript presented in an intelligible fashion and written in standard English?

Reviewer #1: Yes

Reviewer #2: Yes

Reviewer #1: Thank you for inviting me to review this manuscript, please see the attachment for further details.

1.Ensure consistent formatting throughout the manuscript according to the journal guidelines.

2. Provide clearer explanation of corpus design, including selection criteria and data composition.

3. Strengthen connections between literature review and discussion sections, with appropriate citations.

Reviewer #2: 1. The projects funded by the stated sources are unrelated to the current paper.

2. The research methodology and results are not clear. The search period is stated to end on June 1, 2024, but Figure 1 in the results section shows data up to December 2024. Additionally, the sub-corpus of Western English-speaking News Reports on China’s Emergency Medical Assistance to Europe contains 21 valid samples, yet Section 4.3 mentions 23 articles related to "mask diplomacy."

3. The corpus used for Critical Metaphor Discourse Analysis (CEMAE) consists of only 21 samples, which is likely too small to draw conclusions with broad generalizability.

4. The study sample is limited to four Western media outlets, treating "Western media" as a monolithic group and overlooking the potential variations and nuances between different media organizations. The political stance, ownership structure, and target audience of individual media outlets can significantly influence the portrayal of China's image. It is recommended to differentiate between various types of media.

5. The study lacks an interpretation of the cultural and historical context. For instance, while a statistical analysis of the frequency of "mask diplomacy" is provided, the underlying meanings of the data and their connection to Western political and cultural narratives are not sufficiently explored. This is also evident in the analysis of China's dual image in Western media.

6. "Mask diplomacy" is a key concept in the paper; however, it lacks an in-depth exploration of its multifaceted political implications.

**Do you want your identity to be public for this peer review?** For information about this choice, including consent withdrawal, please see our Privacy Policy

Reviewer #1: **Yes: ** Thank you for inviting me to review this manuscript, please see the attachment for further details. Thanks

Reviewer #2: No

---

## [Author Response · Author response to Decision Letter 1]

8 Apr 2025

Dear Editor Rafael Galvão de Almeida,

Thank you very much with your decision letter along with the two reviews of my paper�manuscript # PONE-D-24-60013�. I’ve read your letter and reviews very meticulously. I’d like to show my gratitude for the two profound comments. I understand that I need to make very substantial revisions and I’ve been doing it for a month. I believe I will be able to successfully make all the revisions that the two reviewers suggested because the suggested/requested revisions are all quite specific and detailed even though they will involve a lot of work. An equally important motivation for me to revise the paper is that all two reviewers were very encouraging, as Review #2 commented “The manuscript is technically sound, and the data supports the conclusions” and “The statistical analysis has been performed appropriately and rigorously”. Both the two reviewers agreed “the author has made all data underlying the findings in their manuscript fully available” and “the manuscript presented in an intelligible fashion and written in standard English”. I have been encouraged and inspired by these two reviewers’ comments.

Therefore, I plan to revise my significantly revised paper now. As a result of the extremely substantial revisions I made, the current version of the paper is essentially a new manuscript except that the data and results of the study remain the same. Below I describe in detail how I addressed the issues that you and the reviewers raised the what changed I made. My description of the changes and responses I made in the revision is organized into two main parts. 1)changes/responses addressing the academic editor’s requests and issues and 2)changes/responses addressing the specific comments/suggestions made by each reviewer.

1)Changes/responses addressing the academic editor’s requests and issues

1.Please ensure that your manuscript meets PLOS ONE’s style requirements, including those for file naming.

Response: Thank Editor Rafael Galvão de Almeida for highlighting the importance of adhering to PLOS ONE’s style and formatting requirements. I have revised the manuscript to fully comply with these guidelines, including proper file naming conventions, manuscript structure, and formatting of titles, authors, affiliations, abstract, headings, and references.

“Tianjin Education Commission, China.”

Response: Thank Editor Rafael Galvão de Almeida for his reminder regarding the clarification of the funder’s role. I confirm that the funding agency, Tianjin Education Commission, China, had no involvement in this research. Therefore, I have included this amended “Role of Funder” statement in the cover letter “The funders had no role in study design, data collection and analysis, decision to publish, or preparation of the manuscript.”

“This work was supported by Tianjin Education Commission Research Project, China [grant number 2024SK022];”

“Tianjin Education Commission, China.”

Response: Thank Editor Rafael Galvão de Almeida for pointing out this issue regarding the placement of funding information. In response, I have made the following revisions.I have removed the funding-related text from the Acknowledgments section, as instructed.

Also, I have updated Funding Statement in the Cover Letter: “This work was supported by Tianjin Education Commission Research Project, China [grant number 2024SK022]. The funders had no role in study design, data collection and analysis, decision to publish, or preparation of the manuscript.” Thanks again for your careful review.

2)changes/responses addressing the specific comments/suggestions made by each reviewer.

REVIEWER #1

1.Ensure consistent formatting throughout the manuscript according to the journal guidelines.

Response: Thank Review #1 for his/her valuable feedback. I have thoroughly reviewed my manuscript and revised the formatting to ensure consistency throughout, strictly following the PLOS ONE journal guidelines. Specifically, I have standardized the formatting of headings and subheadings. I have ensured that in-text citations and reference lists adhere to the journal’s citation style. I have adjusted the formatting of figures, tables, and their legends. I have uniformly applied the required font styles, sizes, and line spacing across the manuscript.

I believe these revisions enhance the overall clarity and presentation of my manuscript. Please refer to the revised version for confirmation of these changes.

2. Provide clearer explanation of corpus design, including selection criteria and data composition.

Response: Thank Review #1 for his/her insightful comment regarding the corpus design. In response, I have revised the manuscript to provide a clearer and more detailed explanation of my corpus design, including the selection criteria and data composition. Specifically, I have added additional details in the Methods section to clarify the following:

a. Data Sources and Retrieval Process: I now explain that my corpus was initially constructed using the Factiva database, with articles retrieved using keywords such as “COVID-19,” “China,” and related terms.

b. Inclusion and Exclusion Criteria: I have specified the criteria used for selecting articles, including language (English), publication from reputable Western news outlets, and relevance to the COVID-19 context and China’s role.

c. Corpus Composition: The manuscript now details the composition of my corpus. For example, my primary corpus (CC) comprises 9,783 articles retrieved between December 30, 2019, and June 1, 2024, which was manually screened to yield 9,251 valid articles. A sub-corpus (CHAE) was then created by further filtering these articles using specific keywords (e.g., “China’s emergency humanitarian aid” and “mask diplomacy”), resulting in 22 initial samples and 21 validated articles.

d. Token Counts and Data Scope: I have included token count details and further information on the scope of the data to demonstrate the robustness of our dataset.

e. Rationale for Secondary Searches: Finally, I clarify why secondary searches (using “China’s humanitarian aid”) were conducted within the refined corpus (CC) rather than directly in Factiva, emphasizing the enhanced relevance and specificity of the data.

I believe that these revisions have improved the clarity and transparency of my corpus design and methodology. Thank Review #1 for helping me enhance my manuscript.

3. Strengthen connections between literature review and discussion sections, with appropriate citations.

Response: Thank Review #1 for highlighting the importance of strengthening connections between the literature review and discussion sections. In response to his/her suggestion, I have carefully revised the manuscript to explicitly link key themes and findings in the Discussion back to relevant scholarship reviewed in the Literature Review, incorporating appropriate citations throughout. Specifically, I have made the following modifications:

a. Clarification and Expansion of CDA in the Literature Review : I have explicitly introduced Critical Discourse Analysis (CDA) as a foundational theoretical framework, clearly explaining its relevance and methodological alignment with Critical Metaphor Discourse Analysis (CMDA). Specifically, I included an expanded discussion of CDA’s theoretical principles and its applicability to analyzing metaphorical discourse in media contexts, grounding this discussion in established scholarship (e.g., Charteris-Black, 2004; van Dijk, 1997).

b. Integration of Suggested References: To further enhance theoretical depth, I have incorporated the references Review #1 provided: Lu & Yu (2024), addressing metaphorical discourse in COVID-19 vaccine narratives within Chinese and American media. Yu et al. (2024), focusing on corpus-assisted critical discourse analysis of media representation, illustrating practical applications of CDA. Lu et al. (2022), highlighting image construction through media discourse analysis.

These citations have been strategically positioned to strengthen the theoretical framing in the literature review and directly referenced in the discussion section, reinforcing the coherence and continuity between theoretical foundations and empirical findings.

c. Strengthened Linkages in the Discussion Section:

The Discussion section has been revised to explicitly reference the relevant literature review earlier, thus better aligning empirical observations with theoretical insights and existing scholarship.

For 5.1, I have added:

The findings reinforce previous research indicating that metaphors significantly influence public perceptions by embedding ideological narratives within media discourse (Dong, 2023; Liu, 2023). Consistent with Yang (2023) and Zhang and Lin’s (2023) observations that metaphors serve both persuasive and emotive functions during crises, this study demonstrates how Western media strategically use negatively charged metaphors—such as “manipulative politician” or “dishonest vendor”—to frame China’s humanitarian aid.

For 5.2, I have added:

This study further extends existing scholarship on metaphorical framings. Whereas earlier studies have primarily focused on war, architecture, journey, or body metaphors (Zhang & Lin, 2023; Yang, 2023), our analysis specifically addresses the previously under-examined use of human metaphors. By explicitly attributing human traits to nations, these metaphors carry potent ideological undertones, reflecting and reinforcing particular geopolitical narratives. It supplements Lu and Yu’s (2024) findings, the distinct metaphorical portrayals reflect deeper ideological contrasts and sociocultural contexts, highlighting Western media’s implicit geopolitical biases during the pandemic.

For 5.3, I have added:

The findings presented above explicitly illustrate the frame-building process articulated in framing theory (Goffman, 1974; Gitlin, 2003; Muñiz, 2020). As Beratšová et al. (2018) noted, media framing is influenced by structural factors such as political stance, ownership structure, and demographic variables of target audiences. Consistent with previous research highlighting the use of conflict and negativity frames (Gabore, 2020; Sun, 2021), The Australian’s portrayal illustrates the “China threat” narrative, driven by its conservative ideological stance and corporate ownership, and The Guardian’s ambivalent yet critical stance underscores how editorial independence allows nuanced framing, resonating with its progressive audience. This results support previous framing research by highlighting that media representations of China reflect diverse, strategic editorial choices shaped by complex interactions of ideological, economic, and socio-cultural factors (Lams, 2016; Rodríguez-Wangüemert, 2019; Liu et al., 2022).

Therefore, I believe these revisions substantially improve the clarity and rigor of my manuscript. Thank Review #1 again for his/her constructive comments.

REVIEWER #2

1.The projects funded by the stated sources are unrelated to the current paper.

Response: Thank Review #2 for his/her comment.I confirm that although this study received general support from the Tianjin Education Commission Research Project [Grant number 2024SK022], the funders had no role in study design, data collection and analysis, decision to publish, or preparation of the manuscript. Accordingly, I have revised the Funding Statement to clearly reflect this, in line with PLOS ONE’s policy. The updated Funding Statement now reads:

"This work was supported by Tianjin Education Commission Research Project, China [Grant number 2024SK022]. The funders had no role in study design, data collection and analysis, decision to publish, or preparation of the manuscript."

I hope this clarification is acceptable.

2. The research methodology and results are not clear. The search period is stated to end on June 1, 2024, but Figure 1 in the results section shows data up to December 2024. Additionally, the sub-corpus of Western English-speaking News Reports on China’s Humanitarian aidt to Europe contains 21 valid samples, yet Section 4.3 mentions 23 articles related to "mask diplomacy."

Response: Thank for Review #2 for his/her careful review and for pointing out the inconsistencies in my methodology and results. I appreciate his/her detailed observations. In response, I have made the following revisions:

a. Search Period Discrepancy: I acknowledge the concern regarding the discrepancy between the stated search period and the timeline displayed in Figure 1. To clarify, the data collection was strictly conducted with a search period ending on June 1, 2024, as indicated in the methodology. The discrepancy results from Factiva’s default visualization settings, which automatically present frequency data across standardized annual intervals, causing the x-axis to extend beyond the actual retrieval period. Unfortunately, this visualization format cannot be manually adjusted in Factiva.

To address this and avoid confusion, I have added a clarifying note in the figure caption to explicitly state the actual search period and explain the technical reason for the extended timeline. The underlying data remain accurate and fully consistent with the study design.

b.Sub-corpus Sample Count Inconsistency: I have reviewed my data and confirm that the final sub-corpus comprises 21 valid articles. The mention of 23 articles in Section 4.3 was an oversight. I have updated Section 4.3 to consistently report the correct number of valid samples as 21.

I believe these revisions clarify my methodology and results. Thanks again for his/her valuable feedback, which has helped me improve the transparency and accuracy of my manuscript.

3. The corpus used for Critical Metaphor Discourse Analysis (Sub-corpus) consists of only 21 samples, which is likely too small to draw conclusions with broad generalizability.

Response: Thank Review #2 for his/her comment regarding the size of my corpus for CEMAE. I acknowledge that a corpus of 21 samples may appear limited for drawing conclusions with broad generalizability. However, I would like to emphasize that my study is primarily qualitative in nature. CMDA focuses on an in-depth analysis of linguistic and cognitive features rather than statistical generalizability. The selected 21 articles were carefully curated from reputable Western news sources to capture the specific discourse surrounding China’s Humanitarian aidto Europe during a distinct event. Their rich, focused content provides meaningful insights into the construction of metaphors and the associated cognitive interpretations.

In response to this comment, I have added a discussion of this limitation in the manuscript’s Limitations section. I also clarify that while my findings are based on a limited corpus, they offer valuable, in-depth qualitative insights. Future research with a larger dataset may further validate and expand on my results. I hope these clarifications adequately address your concern.

4. The study sample is limited to four Western media out

---

## [Decision Letter · Decision Letter 1]

Jun 08 2025

Dear Dr. Sun,

We look forward to receiving your revised manuscript.

Kind regards,

Rafael Galvão de Almeida, PhD.

Academic Editor

PLOS ONE

Journal Requirements:

Reviewers' comments:

Reviewer's Responses to Questions

**Comments to the Author**

Reviewer #1: (No Response)

Reviewer #2: (No Response)

2. Is the manuscript technically sound, and do the data support the conclusions?

Reviewer #1: Yes

Reviewer #2: Yes

3. Has the statistical analysis been performed appropriately and rigorously?

Reviewer #1: N/A

Reviewer #2: Yes

4. Have the authors made all data underlying the findings in their manuscript fully available?

Reviewer #1: Yes

Reviewer #2: Yes

5. Is the manuscript presented in an intelligible fashion and written in standard English?

Reviewer #1: Yes

Reviewer #2: Yes

Reviewer #1: Dear Author,

I am so sorry for the delay in providing my feedback.

I am happy to see that the authors have made significant revisions, which have enhanced the overall quality of the paper.

However, upon reviewing the revised manuscript, I noticed a few minor issues, particularly in the usage of certain terminologies. To ensure clarity and consistency throughout the paper, I must provide “minor revision” suggestions. I kindly request that the authors address these small concerns prior to publication. Please see the attachment for detailed information.

Best

Reviewer #2: The research question is not articulated logically enough. The current introduction does not adequately establish clear theoretical or empirical gaps, making it difficult to logically derive the research questions posed. I would suggest that the authors further elaborate on the limitations of previous research or highlight areas that have not been adequately explored in the literature, thus providing a more coherent theoretical rationale for the study. This would help position the research question more convincingly within the existing academic context.

Some parts of the discussion lacked adequate support from the literature. Whilst the interpretation of the findings makes some sense, certain points are made without sufficient citation of relevant research. I would recommend that the authors cite more of the existing literature in their discussion, especially regarding media-specific points. This would enhance the academic rigour and credibility of the discussion.

**Do you want your identity to be public for this peer review?** For information about this choice, including consent withdrawal, please see our Privacy Policy

Reviewer #1: **Yes: ** LU GAOQIANG

Reviewer #2: No

---

## [Author Response · Author response to Decision Letter 2]

25 May 2025

Dear Editor Rafael Galvão de Almeida,

Thank you very much with your decision letter along with the review of my paper�manuscript # PONE-D-24-60013�. I feel encouraged by your comment “it has merit” with a minor revision. I’d like to show my gratitude for the profound comment. Therefore, I plan to revise my paper now meticulously. As a result of the extremely substantial revisions I made, the current version of the paper is the revised manuscript corresponding to the Reviewer’s comment point to point. Below I describe in detail how I addressed the issues that the reviewer raised and what changed I made.

Besides the modifications I’ve made following the Reviewer’s suggestion, I have also carefully revised the sub-section headings to more accurately reflect the empirical content presented, so as to keep the academic rigorous standard. The previous sub-heading “4.1 Reporting Timeline” has been revised to “4.1 Reporting Distribution Over Time”, in order to better describe the changes in reporting intensity throughout the observed period. The previous sub-heading “4.2 The Narrative Behind ‘Mask Diplomacy’” has been revised to “4.2 Frequency and Modifier Analysis of ‘Mask Diplomacy’”, to more precisely capture the focus on lexical patterns and the semantic framing strategies employed by the media. These adjustments aim to enhance the structural transparency and facilitate the readers’ understanding of the empirical results presented.

Followings are my point-to-point reponse to Reviewer#1:

1. Methodology section

a. I noticed that in your manuscript, Section 3.2 is titled “Methodology.” However, “methodology” generally refers to a broader framework that encompasses the overall approach, including data, tools, and analysis. In Section 3.2, you only describe the specific method used in this project. Therefore, I recommend renaming Section 3 as “Methodology” and Section 3.2 as “Method.”

Response: Thank Reviewer#1 for pointing this out. This was indeed an oversight on my part during the previous revision process. As you rightly noted, Section 3 should be titled “Methodology” to reflect the overall research framework, while Section 3.2 should be labeled “Method” to describe the specific procedures used. I have now corrected the section headings accordingly. I appreciate your careful attention to this detail—it has helped improve the manuscript’s clarity and precision. Section 3 is now titled “Methodology” to reflect the overall research design, including corpus construction, analytical tools, and theoretical orientation. Section 3.2 has been renamed “Methods”, focusing specifically on the procedures for identifying and analyzing conceptual metaphors.

b. Additionally, the second paragraph of Section 3.1.1 contains extensive explanations about the criteria for selecting the newspapers. To enhance clarity and conciseness, you might consider presenting that information in a table.

Response: Thank Reviewer#1 for this helpful suggestion. I agree that presenting the selection criteria and background of the newspapers in tabular form would improve both clarity and readability. I have now reorganized this information into a new table in Section 3.1, which summarizes each outlet’s geographic location, political stance, ownership structure, and rationale for inclusion in the corpus.

c. I also suggest combining Sections 3.1.1 (Data Collection) and 3.1.2 (Corpus Construction) since they are closely related and share overlapping content.

Response: Thank Reviewer#1 for the suggestion. I fully understand the rationale for proposing a streamlined structure. However, based on earlier reviewer feedback from a previous round, I was advised to elaborate more clearly and separately on the data collection process and corpus construction procedures, as the earlier description was considered too concise, as Reviwer #1 advised to “Provide clearer explanation of corpus design, including selection criteria and data composition.”.

Furthermore, given that two distinct corpora were developed for this study (Corpus CC and Sub-corpus CHAE), I believe maintaining Sections 3.1.1 and 3.1.2 as separate but logically connected units better reflects the complexity of the research design and improves the manuscript’s clarity. Therefore, I have chosen to retain the current organization while ensuring that the two sections are closely related and avoid redundancy. I sincerely appreciate his/her suggestion, which prompted me to refine the transitions between the two subsections for better coherence.

d. Moreover, some details within the current methodology section appear to be better suited for the “Findings” section. Please consider relocating this content accordingly.

Response: Thank Reviewer#1 for pointing this out. Upon review, I agree that certain descriptive elements—particularly those involving preliminary observations and early data patterns—are more appropriate for the Findings section. I have revised the Methododology section to ensure that it focuses solely on describing the data sources, corpus construction, and analytic procedures. Content that involved preliminary interpretation of findings, such as the identification of THE NATION AS PERSON and its secondary mappings (e.g., CHINA AS PERSON, EUROPE AS PERSON), has been relocated to the Findings section. Now, the Methodology section exclusively outlines the procedures without reporting research outcomes, in line with academic writing conventions. I sincerely appreciate his/her suggestion to improve the academic rigor of the manuscript.

e. Finally, the manuscript would benefit from a dedicated data analysis section that demonstrates, for example, showing your readers an example about how metaphor(s) is identified in the dataset (i.e., MIP or MIPVU).

Response: Thank Reviewer#1 for this constructive suggestion. I fully agree that providing a concrete example of the metaphor identification process enhances the transparency and replicability of the study. In the revised manuscript, I have, as the reviewer suggested, added a dedicated Data Analysis section (Section 3.3) that systematically describes the procedure based on the MIPVU guidelines. Moreover, I included a detailed example from The Guardian (April 29, 2020), where the expression “fan mistrust” was analyzed to illustrate how contextual and basic meanings were compared to determine the metaphorical status. I believe this addition strengthens the methodological rigor and clarity of the paper.

f. Based on these points, I recommend restructuring the methodology section as follows:

3. Methodology

3.1 Data

3.2 Methods

3.3 Data Analysis

I hope these suggestions help improve the clarity and structure of your manuscript.

Response: Thank Reviewer#1 very much for your rigorous and constructive feedback. I do benefit tremendously with the recommendation of structuing the methodology section. Hence, I have revised the structure of the Methodology section accordingly. The section is now organized into three subsections: 3.1 Data, 3.2 Methods, and 3.3 Data Analysis. In addition, I have expanded subsection 3.3 to explicitly describe the basic corpus statistics and the metaphor identification procedures, thereby enhancing the transparency and clarity of the methodological design. I sincerely appreciate his/her valuable suggestions, which have helped to improve the overall coherence and academic rigor of the manuscript.

2. Terminology issues:

a. AntConc

In the quantitative analysis section, you refer to AntConc 3.5.9 as a vocabulary analysis software. However, AntConc is more described as a “corpus analysis toolkit” (please also cite from Anthony). I recommend updating the description to reflect its true functionality.

Response: Thank Reviewer for highlighting this important terminology issue. I have revised the description of AntConc in the Methods section to accurately refer to it as a “corpus analysis toolkit”, following the terminology used by Anthony (2023). The relevant citation has also been added to properly acknowledge the source. Thanks again for his/her valuable and meticulous correction, which has helped me improve the academic accuracy of my manuscript.

b. MIP vs. MIPVU

In Section 3.2, the manuscript states that the MIP method was utilized in this project, while in Section 4.4, a different method, MIPVU, is mentioned. Please ensure consistency in the naming of the method throughout the manuscript and cite the related reference for the chosen method.

Response: Thank Reviewer for drawing my attention to this inconsistency. I have carefully reviewed the manuscript and corrected all mentions to consistently refer to the MIPVU method throughout the text. In this study, the Metaphor Identification Procedure Vrije Universiteit (MIPVU) (Steen et al., 2010) was adopted to identify metaphorical expressions within the corpus. Compared to the original Metaphor Identification Procedure (MIP) proposed by the Pragglejaz Group (2007), MIPVU offers a more detailed and standardized framework for linguistic metaphor identification, particularly suitable for large and complex datasets such as media discourse. Its explicit coding conventions and expanded categorization criteria enhance the transparency, consistency, and replicability of metaphor analysis, aligning with the requirements of corpus-based critical discourse studies.

Additionally, I have included the appropriate citation for MIPVU, following Steen et al. (2010), in both the Methods and Data Analysis sections, and added the corresponding reference to the References list. Thanks again for his/her valuable feedback, which has helped me improve the accuracy terminology in my manuscript.

c. Section 4 (Results)

Please note the essential difference between “Results” and “Findings”:

The former one should present the empirically obtained data, including statistical analyses, charts, and other measurable experimental indicators, whereas the later one is intended for interpreting, summarizing, and analyzing these data.

Response: Thank Reviewer for his/her thoughtful comment regarding the distinction between “Results” and “Findings”. I appreciate his/her academic explanation. In response, I have made the following revisions. Upon reflections, I recognized that parts of the section not only presented empirical data but also included preliminary analysis and summarization of the findings based on the data. Therefore, to more accurately reflect the content and maintain terminological precision, following the Reviewer’s suggetion, I have revised the section title to “Findings” in the updated manuscript. I hope this adjustment clarifies the structure and better to follow academic conventions.

Followings are my point-to-point reponse to Reviewer#2:

Reviewer #2: The research question is not articulated logically enough. The current introduction does not adequately establish clear theoretical or empirical gaps, making it difficult to logically derive the research questions posed. I would suggest that the authors further elaborate on the limitations of previous research or highlight areas that have not been adequately explored in the literature, thus providing a more coherent theoretical rationale for the study. This would help position the research question more convincingly within the existing academic context.

Response: Thank Reviewer #2 for this valuable suggestion. I agree that the original introduction could be improved in its articulation of the theoretical and empirical gaps. In the revised manuscript, I have thoroughly revised the Introduction and Literature Review sections to address this issue. Specifically, I now clearly outline three underexplored aspects in the existing literature: (1) the lack of corpus-based analyses of Western media coverage of China’s humanitarian aid during COVID-19; (2) the absence of research combining both quantitative corpus techniques and qualitative metaphor analysis; and (3) the under-examined ideological implications of the label “mask diplomacy.” These gaps are now explicitly presented in Section 1, final paragraph (pp. 2–3), leading logically to the two research questions. I hope this revision provides a more coherent and convincing rationale for the study.

Some parts of the discussion lacked adequate support from the literature. Whilst the interpretation of the findings makes some sense, certain points are made without sufficient citation of relevant research. I would recommend that the authors cite more of the existing literature in their discussion, especially regarding media-specific points. This would enhance the academic rigour and credibility of the discussion.

Response: I sincerely thank the reviewer#2 for this insightful and constructive suggestion. In response, I undertook a substantial revision of the Discussion section to address the issue of insufficient literature support and to improve overall conceptual clarity and depth.

Firstly�I have now restructured the main discussion part into three distinct subsections (5.1, 5.2, 5.3 and 5.4) in order to better differentiate between lexical strategies, metaphorical construction, media-specific ideological framing, and different media positioning. This restructuring not only improves the clarity and logical flow but also enables us to better align each interpretive dimension with appropriate theoretical and empirical literature.

Originally, the discussion consisted of: 5.1. China’s image in western media’s discursive strategies in THE NATION AS PERSON; 5.2. Ideological implication of THE NATION AS PERSON;5.3. Media outlet’s positionality and its framing of nation’s image. However, I found that the original Section 5.1 was internally heterogeneous, discussing both the discursive use of “mask diplomacy” and the metaphorical construction of China’s image under the “nation-as-person” schema. This not only blurred the topical focus of that section but also led to significant conceptual overlap with Section 5.2.

Secondly, I have significantly strengthened the discussion section by integrating a wider range of relevant academic literature to better support our interpretations and clarify the theoretical positioning of our findings. The following modifications have been made to improve academic rigour:

5.1. Lexical Framing of “Mask Diplomacy”

• Strengthened theoretical grounding by referencing Lakoff & Johnson’s (1980) Conceptual Metaphor Theory and Charteris-Black’s (2004) Critical Metaphor Analysis.

• Responded to prior research on “mask diplomacy”, including Kowalski (2021), Fahrisa et al. (2023), and Müller et al. (2024). I clarified that while these studies emphasize geopolitical and diplomatic interpretations, my contribution lies in examining how the term is linguistically operationalized in Western news discourse.

• Contribution clarified: I positioned my study as an extension of these works, offering a corpus-based discourse-linguistic perspective that highlights how evaluative language and metaphorical choices function ideologically.

5.2. China’s image in western media’s discursive strategies in THE NATION AS PERSON

• Broadened literature context by incorporating work on metaphor in media discourse (e.g., Charteris-Black, 2004; Zhang & Lin, 2023).

• Clarified thematic novelty: I explained that while many metaphor studies focus on war, journey, or spatial metaphors, our study foregrounds personification metaphors such as “CHINA AS POLITICIAN” and “AMERICA AS HOTHEAD.”

5.3. Media Outlet’s Positionality and Framing

• Supported claims with empirical literature, including studies on media ownership, ideology, and geopolitical bias [10], [12], [21], [25], [30], [35], [38].

• Highlighted contribution: I explained how my comparative analysis of The New York Times, The Guardian, and The Australian provides a fine-grained account of metaphor variation across outlets—showing how the same metaphor (e.g., CHINA AS POLITICIAN) is differently realized depending on political stance, ownership, and target readership.

In summary, I have expanded the discussion from two to three structured subsections; enhanced each part with relevant theoretical and empirical citations;clarified how this study extends and enriches prior research; improved the overall academic rigour and interpretive precision of the manuscript.

I am grateful for the reviewe

---

## [Editor Report · Decision Letter 2]

On China’s image constructed from western news coverage of China’s humanitarian aid

PONE-D-24-60013R2

Dear Dr. Sun,

We’re pleased to inform you that your manuscript has been judged scientifically suitable for publication and will be formally accepted for publication once it meets all outstanding technical requirements.

Kind regards,

Rafael Galvão de Almeida, PhD.

Academic Editor

PLOS ONE

Additional Editor Comments (optional):

Correct a typo in "as it has more preference offically and academically,"

In "Moreover, in the context of the China-US trade war, strategic framing is more evident in coverage from countries directly involved in the conflict, underscoring the influence of geopolitical interests on framing choices", although the manuscript was initially submitted before Trump started his tariff policies, a reference to them could be interesting to add

"Despite of its misinterpretation of this Chinese proverb" - what is the usual interpretation of the proverb? You can add in the text or in a footnote

The manuscript has the message "Error! Reference source not found" throughout the text, please correct them
---

## [Editor Report · Acceptance letter]

PONE-D-24-60013R2

PLOS ONE

Dear Dr. Sun,

I'm pleased to inform you that your manuscript has been deemed suitable for publication in PLOS ONE. Congratulations! Your manuscript is now being handed over to our production team.

Kind regards,

on behalf of

Dr. Rafael Galvão de Almeida

Academic Editor

PLOS ONE